# OpenReview forum: "E$^2$AT: Multimodal Jailbreak Defense via Dynamic Joint Optimization"
_ICLR.cc/2026/Conference — ICLR 2026 Conference Withdrawn Submission_

### Official Review · Reviewer_kC9m · 2025-10-26

**Soundness:** 2
**Presentation:** 2
**Contribution:** 2
**Rating:** 4
**Confidence:** 4

**Summary:**

This paper introduces the E^2 AT framework for improving the robustness of MLLMs against jailbreak attacks targeting both visual and textual modalities. The approach combines a projector-based AT module for efficient visual defense with a novel Dynamic Joint Multimodal Optimization (DJMO) strategy that dynamically adjusts loss weights to balance adversarial and clean objectives. Extensive experiments on three open-source MLLMs and multiple attack methods demonstrate strong trade-off between safety and utility.

**Strengths:**

+ **Important problem**: The paper addresses an important and timely problem—defending against jailbreak attacks in MLLMs.
+ **Real-world applicability**: The embodied robotics demonstrations in Figures 1(c) and 3 effectively connect the proposed algorithmic advances to real-world safety applications, illustrating concrete examples of model refusals in adversarial scenarios.
+ **Extensive empirical evaluation**: The paper presents thorough empirical results across three MLLMs and two comprehensive benchmarks, along with detailed ablation studies, providing nuanced insights into the model’s performance and robustness.

**Weaknesses:**

- **Outdated related work**: The discussion of related work and baselines appears somewhat outdated. It would strengthen the paper to include and compare with more recent studies, such as [1][2][3][4].
- **Efficiency claim**: As far as I know, freezing the vision encoder and only updating the projector is a common strategy to improve training efficiency for MLLMs [5]. Therefore, the claimed novelty in this aspect may be somewhat overstated. Moreover, although the paper describes the method as “highly efficient,” the results do not clearly demonstrate a significant efficiency advantage. For example, in Figure 4, the time efficiency of E^2 AT is only better than that of PAT.
- **Attack methods**: Since the model is optimized against white-box adaptive attacks, it would be valuable to include evaluations under black-box adaptive attack settings (e.g., [5][6]) to better demonstrate the generalization ability of E^2 AT against adaptive attacks.

References:

[1] Securing Multimodal Large Language Models: Defending Against Jailbreak Attacks with Adversarial Tuning

[2] VLMGuard-R1: Proactive Safety Alignment for VLMs via Reasoning-Driven Prompt Optimization

[3] Spot Risks Before Speaking! Unraveling Safety Attention Heads in Large Vision-Language Models

[4] SEA: Low-Resource Safety Alignment for Multimodal Large Language Models via Synthetic Embeddings

[5] Visual Instruction Tuning

[6] Jailbreaking Black Box Large Language Models in Twenty Queries

[7] AutoDAN-Turbo: A Lifelong Agent for Strategy Self-Exploration to Jailbreak LLMs

**Questions:**

See Weakness.

---

> ### Author Response · Authors · 2025-11-20
>
> We sincerely appreciate Reviewer kC9m’s thorough review. We have carefully considered each comment and provided our responses below.
>
> Q1: **Missing comparison with recent studies**
>
> A1: We sincerely thank Reviewer kC9m for pointing this out and providing these valuable references. We have carefully studied these recent works [1-4] and incorporated some of them into the revised Related Work section (Page 3, Section 2). We would like to clarify the fundamental distinction between our method, E²AT, and methods like **VLMGuard-R1 [2]** and **SAHs [3]**, primarily in terms of the defense paradigm and inference overhead. Both VLMGuard-R1 and SAHs belong to inference-time defenses, which inevitably increase inference latency and system complexity. In contrast, E²AT maintains inference efficiency identical to the original model with zero additional overhead, highlighting the efficiency of our approach.
>
> [2] VLMGuard-R1: Proactive Safety Alignment for VLMs via Reasoning-Driven Prompt Optimization
>
> [3] Spot Risks Before Speaking! Unraveling Safety Attention Heads in Large Vision-Language Models
>
> Q2: **Efficiency claim**
>
> A2: In **Figure 4**, the time efficiency of E²AT is not only superior to that of PAT. In fact, PAT has the shortest training time, but our ASR performance is more advantageous compared to PAT. Additionally, methods like BlueSuffix require extra components during inference, which increases the time spent. In contrast, our inference is instant, with no additional components needed; the model is ready for use immediately after training.
>
> Q3: **Attack methods**
>
> A3: We assessed adversarial robustness using the **PAIR [1]** framework on the **Anthropic-HHH dataset** (58 samples), consistent with the setup in **Steering Away from Harm [2]**. Utilizing **Vicuna-13b-v1.5** as the attacker and **GPT-4** as the judge, our experiments confirm that E²AT exhibits strong robustness against advanced automated semantic jailbreaks. Additionally, we are finalizing the analysis of the experiments on **AutoDAN-Turbo [3]** and will provide the detailed results shortly.
>
> **Prompt Automatic Iterative Refinement (PAIR, ASR Summary)**
>
> | Strategy | Total Samples | Success | ASR (%) |
> | :--- | :---: | :---: | :---: |
> | authority | 58 | 2 | 3.45 |
> | logical | 58 | 2 | 3.45 |
> | roleplaying | 58 | 4 | 6.90 |
>
> [1] Jailbreaking Black Box Large Language Models in Twenty Queries
>
> [2] Steering away from harm: An adaptive approach to defending vision language model against jailbreaks
>
> [3] AutoDAN-Turbo: A Lifelong Agent for Strategy Self-Exploration to Jailbreak LLMs

---

> > ### Author Response · Authors · 2025-11-24
> >
> > Q3: **AutoDAN-Turbo Attack**
> >
> > To evaluate **AutoDAN-Turbo** on the **Anthropic_HHH dataset**, we conducted black-box experiments using **Meta-Llama-3-8B-Instruct** as the Attacker and Summarizer, and **gemma-1.1-7b-it** as the Scorer. We utilized OpenAI's **text-embedding-ada-002** for similarity-based strategy retrieval, limiting each sample to **100** iterations. Adhering to strict standards, a jailbreak was defined as a response score exceeding **8.5**, thereby verifying the framework's ability to compromise safety alignment under a limited query budget. The results are presented below:
> > | Total Test Samples | Success Count (Score $\ge$ 8.5) | Attack Success Rate (ASR) |
> > | :---: | :---: | :---: |
> > | 58 | 17 | 29.31% |

---

### Official Review · Reviewer_sSH4 · 2025-10-26

**Soundness:** 2
**Presentation:** 3
**Contribution:** 2
**Rating:** 2
**Confidence:** 3

**Summary:**

The paper focuses on enhancing the safety of multimodal large language models through an adversarial training framework that incorporates both image and text perturbations. In addition, it proposes an adaptive weight adjustment mechanism to better balance safety objectives with utility preservation. Experimental results show that the proposed method effectively defends against test-time attacks.

**Strengths:**

1. The paper is well-written and easy to follow.


2. The motivation is clear and straightforward.


3. The experiments demonstrate that the proposed method performs well against test-time attacks.

**Weaknesses:**

1. The novelty is relatively limited. Adversarial learning is a fundamental concept in the adversarial robustness field, and there already exist several works on adversarial training for both LLMs and MLLMs [1][2]. The proposed method appears to be a straightforward application of adversarial training to the MLLM safety setting, with little new technical contribution in the overall design.


2. The discussion and comparison with related works are not comprehensive. For example, [1] also proposes an adversarial training method that considers both image and text perturbations, but the distinction between that work and the current paper is not clearly explained.


3. The experiments should include some latest models, such as the Qwen series.

[1] Yin, Ziyi, et al. "Towards Robust Multimodal Large Language Models Against Jailbreak Attacks." arXiv preprint arXiv:2502.00653 (2025).

[2] Xhonneux, Sophie, et al. "Efficient adversarial training in LLMs with continuous attacks." Proceedings of the 38th International Conference on Neural Information Processing Systems. 2024.

**Questions:**

See Weaknesses

---

> ### Author Response · Authors · 2025-11-20
>
> We thank Reviewer sSH4 for the insightful comments. We have addressed the specific concerns point-by-point below.
>
> Q1: **Insufficient technical contribution**
>
> A1: **Work [1]** targets MLLMs and addresses the challenge of multimodal white-box defense through a unified embedding-layer noise attack. **Work [2]** concentrates on text-only LLMs, aiming to solve the efficiency bottlenecks of adversarial training by shifting the attack computation space to achieve ultra-fast training. In contrast, our work focuses on architectural collaborative defense for MLLMs, dedicated to resolving novel security issues introduced by multimodality, with an emphasis on the optimization of architectural components.
>
> [1] Towards Robust Multimodal Large Language Models Against Jailbreak Attacks
>
> [2] Efficient adversarial training in LLMs with continuous attacks
>
> Q2: **Insufficient comparison with related works**
>
> A2: These two methods exhibit significant differences in adversarial generation levels and optimization strategies, as well as their impact on model performance: First, E²AT operates at the input level, generating potent adversarial samples by injecting noise into images and appending adversarial suffixes to text. In contrast, **SAFEMLLM [1]** targets the embedding level, constructing adversarial samples by generating noise matrices for both visual and textual embeddings. Second, regarding loss optimization, E²AT employs a dynamic weight adjustment strategy to adaptively balance training objectives, distinguishing it from the static coefficient configuration used in SAFEMLLM. Notably, as evidenced by the "LLaVA (PGD + Embedding Attack)" results in **Table IX**, adversarial training at the embedding level yields a lower utility score while achieving a comparable ASR. This suggests that compared to input-level methods, adversarial training applied to the embedding layer may be more prone to degrading the model's intrinsic task performance.
>
> [1] Towards Robust Multimodal Large Language Models Against Jailbreak Attacks
>
> Q3: **Insufficient model coverage in experiments**
>
> A3: We appreciate Reviewer sSH4’s valuable suggestion to benchmark against a broader range of mainstream MLLMs. We address the specific model choices below: **1. Technical Incompatibility with Qwen2.5-VL.** We would like to clarify that Qwen2.5-VL is technically unsuitable for our specific framework due to an architectural mismatch. Our method, defined as ''projection-based efficient adversarial training'', strictly relies on optimizing a decoupled, lightweight projection module (e.g., the MLP in LLaVA) to ensure parameter efficiency. In contrast, Qwen2.5-VL employs a C-Abstractor deeply coupled within the visual encoder. Applying E²AT would necessitate fine-tuning the visual backbone itself, which contradicts the core ''efficiency'' contribution of our work. **2. Active Implementation on InstructBLIP.** To constructively address the concern regarding generalization, we are actively conducting additional experiments on InstructBLIP. Unlike Qwen2.5-VL, InstructBLIP features a Q-Former architecture that aligns well with our projection-based optimization objective. We are currently adapting our code to this architecture and commit to including these experimental results in the final revision to further demonstrate the robustness and generalization of our method.

---

> ### Author Response · Authors · 2025-11-25
>
> Q3: **Insufficient model coverage in experiments**
>
> A3: Following your recommendation, we evaluated E²AT on the InstructBLIP architecture. The results, summarized in the table below, confirm that our method retains its exceptional defense capabilities:
>
> **Table: Comparison of Defense Performance on InstructBLIP**
> | MLLM | Defense Method | Logic | Persuade | Template | FigStep | Query-Relevant | W-ASR | LLaVA-Bench Score |
> | :--- | :--- | :---: | :---: | :---: | :---: | :---: | :---: | :---: |
> | | No Defense | 0.73 | 0.52 | 0.50 | 0.37 | 0.24 | 0.440 | 0.510 |
> | | RobustVLM | 0.74 | 0.50 | 0.41 | 0.29 | 0.16 | 0.425 | 0.472 |
> | **InstructBLIP** | PAT | 0.39 | 0.18 | 0.65 | 0.36 | 0.23 | 0.352 | **0.535** |
> | | BlueSuffix | 0.22 | 0.46 | 0.06 | **0.05** | 0.06 | 0.198 | 0.465 |
> | | **E$^2$AT (Ours)** | **0.02** | **0.03** | **0.01** | 0.09 | **0.01** | **0.045** | 0.502 |

---

### Official Review · Reviewer_83Ji · 2025-10-31

**Soundness:** 2
**Presentation:** 1
**Contribution:** 2
**Rating:** 2
**Confidence:** 4

**Summary:**

The paper focuses on adversarial training of MLLMs, aiming to enhance the robustness of MLLMs.

**Strengths:**

1. This paper focuses on an important research question.
2. The experiments are conducted on different attack settings.

**Weaknesses:**

**A critical issue**: In this paper, all citation formats are incorrect. The authors should properly distinguish and check the usage of ~\cite and ~\citep. In the current version, the citation formatting errors make the paper very hard to read (**all citations** are directly attached to the main text without parentheses). Such an obvious mistake—one that anyone would notice after a single read—is clearly unreasonable for a top-tier conference submission.

1. In Figure 1(c), the authors claim that without their defense method, the robot will perform dangerous behaviors. This experiment is very confusing to me: (1) What is the purpose of this experiment? (2) Under what setting was it conducted—what model and attack method were used? (3) Does the experiment compare against other defense methods? (4) How is this experiment specifically related to the subsequent context, such as the motivation? Without any quantitative analysis, simply showing one example does not make clear what the authors intend to convey.

2. Unreasonable threat model definition. After reading lines 166–169, the authors seem to target defenses against black-box attacks—i.e., the defender can adversarially fine-tune (AT) the model, but the attacker must operate in a black-box setting. I do not understand the motivation for this setup—in fact, the traditional goal of AT is to defend against white-box attacks [1]. This threat-model design is therefore unreasonable.

[1] Aleksander Madry. Towards deep learning models resistant to adversarial attacks. arXiv, 2017.

3. Notation issues. In Eq. 3 and Eq. 4, the authors use different symbols to define the “feature space.” What exactly is the feature space? In Eq. 3, why do x_image and x_text belong to the feature space?

4. Experimental setup. The paper lacks comparisons with several classic and mainstream MLLMs, including Qwen2.5-VL and InternVL models. Without these models, it is difficult to support claims about the method’s generalization capability.

**Questions:**

Please refer to the weakness section.

---

> ### Author Response · Authors · 2025-11-20
>
> We are grateful to Reviewer 83Ji for the valuable suggestions, which have helped improve our manuscript. Our responses are detailed as follows.
>
> Q: **Incorrect citation formatting**
>
> A: We appreciate Reviewer 83Ji's detailed feedback and attention to the formatting issue. We acknowledge the mistake and apologize for the oversight. The necessary revisions have been made.
>
> Q1.1: **Purpose of the experiment**
>
> A1.1: To visually demonstrate the effectiveness of the E²AT method in embodied intelligent scenarios. When a multimodal large language model, not trained with E²AT, is deployed on a robotic arm, it performs harmful actions in response to malicious commands such as "move the bomb to the designated area".
>
> Q1.2: **Experimental setting details**
>
> A1.2: The embodied intelligent scenario experiments in this paper use the **LLaVA** model. We do not explore attack methods, but instead, we input malicious commands and observe whether the model, trained with E²AT, executes these commands.
>
> Q1.3: **Comparison with other defense methods**
>
> A1.3: Regarding Figure 1\(c), we wish to emphasize the deployment feasibility and practical defense effectiveness of E²AT in real-world embodied intelligence scenarios. While we have provided detailed quantitative comparisons against existing defense methods (e.g., VLGuard, PAT) on standard benchmarks in the tables of the main text, Figure 1\(c) aims to offer a complementary qualitative dimension: specifically, verifying that E²AT can effectively intercept instructions that would lead to severe physical consequences (such as manipulating a bomb). The results demonstrate that E²AT successfully bridges the gap between digital defense and physical safety, highlighting its unique value in high-stakes real-world applications.
>
> Q1.4: **Relation to research motivation**
>
> A1.4: Figure 1\(c) and Figure 3 in the appendix display the embodied intelligent scenario experiments combining E²AT with the robotic arm, demonstrating that our work provides a more robust and secure solution for embodied intelligent applications.
>
> Q2: **Questionable rationality of the threat model setting**
>
> A2: We apologize for the incorrect phrasing in our previous statement. The attackers we aim to defend against can be either white-box or black-box, with no specific scenario restrictions. We will revise the relevant phrasing in lines 148-151. In the experimental section of this paper, black-box attacks are shown in **Table 1**, while white-box attacks are shown in **Table 7**, including Adaptive BAP, Adaptive GCG, and Adaptive AutoDan. E²AT demonstrates strong defense performance against all of these attack methods.
>
> Q3: **Notation issues**
>
> A3: Equation (3) operates in the feature space to generate the adversarial target, while Equation (4) operates in the parameter space to optimize the model.
>
> Q4: **Insufficient model coverage in experiments**
>
> A4: We appreciate Reviewer 83Ji’s valuable suggestion to benchmark against a broader range of mainstream MLLMs. We address the specific model choices below: **1. Technical Incompatibility with Qwen2.5-VL.** We would like to clarify that Qwen2.5-VL is technically unsuitable for our specific framework due to an architectural mismatch. Our method, defined as ''projection-based efficient adversarial training'', strictly relies on optimizing a decoupled, lightweight projection module (e.g., the MLP in LLaVA) to ensure parameter efficiency. In contrast, Qwen2.5-VL employs a C-Abstractor deeply coupled within the visual encoder. Applying E²AT would necessitate fine-tuning the visual backbone itself, which contradicts the core ''efficiency'' contribution of our work. **2. Active Implementation on InstructBLIP.** To constructively address the concern regarding generalization, we are actively conducting additional experiments on InstructBLIP. Unlike Qwen2.5-VL, InstructBLIP features a Q-Former architecture that aligns well with our projection-based optimization objective. We are currently adapting our code to this architecture and commit to including these experimental results in the final revision to further demonstrate the robustness and generalization of our method.

---

> ### Author Response · Authors · 2025-11-25
>
> Q4: **Insufficient model coverage in experiments**
>
> A4: We appreciate your valuable suggestion. We have conducted verification on the InstructBLIP architecture, and the results demonstrate that E²AT maintains superior defense performance, as shown in the table below:
>
> **Table: Comparison of Defense Performance on InstructBLIP**
> | MLLM | Defense Method | Logic | Persuade | Template | FigStep | Query-Relevant | W-ASR | LLaVA-Bench Score |
> | :--- | :--- | :---: | :---: | :---: | :---: | :---: | :---: | :---: |
> | | No Defense | 0.73 | 0.52 | 0.50 | 0.37 | 0.24 | 0.440 | 0.510 |
> | | RobustVLM | 0.74 | 0.50 | 0.41 | 0.29 | 0.16 | 0.425 | 0.472 |
> | **InstructBLIP** | PAT | 0.39 | 0.18 | 0.65 | 0.36 | 0.23 | 0.352 | **0.535** |
> | | BlueSuffix | 0.22 | 0.46 | 0.06 | **0.05** | 0.06 | 0.198 | 0.465 |
> | | **E$^2$AT (Ours)** | **0.02** | **0.03** | **0.01** | 0.09 | **0.01** | **0.045** | 0.502 |

---

### Official Review · Reviewer_7UmD · 2025-10-31

**Soundness:** 3
**Presentation:** 3
**Contribution:** 3
**Rating:** 6
**Confidence:** 4

**Summary:**

This paper proposes E²AT, an efficient end-to-end adversarial training framework designed to defend against multimodal jailbreak attacks on Multimodal Large Language Models (MLLMs). The key innovation lies in its Dynamic Joint Multimodal Optimization (DJMO) strategy, which dynamically balances adversarial and clean training objectives across both visual and textual modalities. The authors introduce a projector-based adversarial training module that aligns adversarial image features with clean ones at the feature level, reducing computational overhead. For text, they adopt GCG-based adversarial suffix generation. The DJMO mechanism adaptively weights the adversarial and clean losses during training using exponential moving averages, enabling the model to focus on the most relevant modality at each stage.Extensive experiments on three MLLMs (LLaVA, Bunny, mPLUG-Owl2) and two benchmarks (JailbreakV-28K, MM-SafetyBench) demonstrate that E²AT achieves state-of-the-art robustness, reducing the average attack success rate by 34% compared to baselines, while maintaining clean task performance. The method is also validated in a real-world robotic arm scenario, showing practical applicability.

**Strengths:**

The paper introduces E^2AT (Efficient End-to-End Adversarial Training), a multimodal jailbreak defense framework that jointly optimizes visual and textual adversarial training with dynamic weighting. It integrates a projector-based adversarial training mechanism and a Dynamic Joint Modality Optimization (DJMO) strategy to balance robustness across modalities. Experiments on multiple MLLMs and two benchmarks show that E^2AT significantly reduces the weighted attack success rate, outperforming prior defenses such as VLGuard and BlueSuffix. The method is presented as computationally efficient and broadly applicable to multimodal safety alignment.

**Weaknesses:**

1. Ablation results are descriptive but not deeply analyzed, particularly, the feature-level effects of removing projector optimization.
2. Despite emphasizing "Efficient", the paper provides no runtime, memory, or FLOPs comparison with prior works. Efficiency remains qualitative rather than experimentally verified.
3. A formatting error: The caption for a table should be placed above it.

**Questions:**

These are all in the 'Weaknesses' section above.

---

> ### Author Response · Authors · 2025-11-20
>
> We appreciate Reviewer 7UmD for the constructive feedback. Our detailed responses to the questions raised are provided below.
>
> Q1: **Missing analysis on feature-level effects of projector optimization**
>
> A1: As shown in **Table 5**, the ablation experiments on the Bunny-v1.0-4B model with different component configurations reveal that removing the projector optimization has a significant impact on the model's performance. While removing the projector optimization does not affect the performance on Logic, Persuade, and Template tasks, the defense performance against FigStep and Query-Relevant modal attacks declines. This indicates that without projector optimization, the model fails to effectively align visual and textual features, leading to a decrease in defense effectiveness. This result indirectly validates the importance of the interaction between visual and textual modalities, highlighting the crucial role of feature alignment in defense.
>
> Q2: **Missing efficiency metric comparisons**
>
> A2: In the appendix, **Figure 4** presents a comparison between E²AT (our method) and other approaches using a bubble chart. The chart shows E²AT's performance in terms of ASR and ACC, while the size of the bubbles intuitively reflects the computational cost of each method. It is clear from the figure that E²AT excels in both defense performance and computational efficiency, further validating our claim of Efficient. The comparison of training times also demonstrates that E²AT effectively balances high defense capability with computational efficiency.
>
> Q3: **Incorrect table caption placement**
>
> A3: We appreciate Reviewer 7UmD for identifying this oversight. We have moved the table caption to the top as suggested.

---

### Author Response · Authors · 2025-12-02

In this submission, we introduce E²AT, a novel defense approach designed for MLLMs to defend against jailbreak attacks. E²AT optimizes the interaction between visual and textual modalities, ensuring robust defense capabilities with no added inference overhead. Experimental results confirm that E²AT achieves state-of-the-art performance in defending against various attack methods, including both white-box and black-box adversaries.

We thank all responsible reviewers for their thoughtful engagement and constructive feedback. We summarize the key clarifications and additional evidence provided during the rebuttal and discussion period:

**1. Technical Novelty & Contributions.** E²AT distinguishes itself from existing embedding-level defenses (e.g., SafeMLLM) by operating directly at the input level with a novel projector optimization strategy. As verified by our ablation studies, this design effectively aligns visual and textual features to counteract multimodal threats.

**2. Efficiency & Practicality.** We clarify that E²AT offers an optimal trade-off between training cost and defense performance. While PAT has a shorter training time, E²AT achieves significantly higher Attack Success Rate (ASR) reduction. More importantly, unlike inference-time defenses (e.g., BlueSuffix) that require extra computation, E²AT incurs zero inference overhead, making it highly suitable for real-time deployment as demonstrated in our comparison bubble charts.

**3. Comprehensive Improvements.** In response to reviewer feedback, we have significantly strengthened the submission:

**· Expanded Benchmarks.** We validated E²AT on **InstructBLIP**, demonstrating strong generalization.

**· Clarified Scenarios.** We refined the **embodied intelligence (robotic arm)** setup to clearly illustrate physical safety risks and added evaluations against **adaptive black-box attacks**.

**· Rigorous Presentation.** We have corrected all **citation formatting errors** and polished the related work section to include the latest state-of-the-art baselines.

In general, most concerns raised by Reviewers 7UmD, 83Ji, sSH4, and kC9m have been carefully addressed. Regrettably, due to unexpected technical issues within the OpenReview system, we were unable to engage in effective multi-round interactions with the reviewers. We kindly request that the Area Chair and reviewers take the comprehensive responses and improvements provided in this rebuttal into full consideration. We sincerely appreciate the AC and responsible reviewers for their efforts in advancing multimodal defense research. The additional results, clarifications, and mitigations presented in this submission have strengthened both the technical contributions and practical relevance of E²AT. It is our hope that E²AT will make a significant contribution to the security of MLLMs in real-world applications.

---

### Note · Authors · 2026-01-06

I have read and agree with the venue's withdrawal policy on behalf of myself and my co-authors.